# The Function, Regulation, and Mechanism of Protein Turnover in Circadian Systems in *Neurospora* and Other Species

**DOI:** 10.3390/ijms25052574

**Published:** 2024-02-22

**Authors:** Haoran Zhang, Zengxuan Zhou, Jinhu Guo

**Affiliations:** MOE Key Laboratory of Gene Function and Regulation, School of Life Sciences, Sun Yat-sen University, Guangzhou 510275, China; zhanghr83@mail2.sysu.edu.cn (H.Z.); zhoucx6@mail2.sysu.edu.cn (Z.Z.)

**Keywords:** circadian clock, protein stability, post-translational control, circadian period, phosphorylation

## Abstract

Circadian clocks drive a large array of physiological and behavioral activities. At the molecular level, circadian clocks are composed of positive and negative elements that form core oscillators generating the basic circadian rhythms. Over the course of the circadian period, circadian negative proteins undergo progressive hyperphosphorylation and eventually degrade, and their stability is finely controlled by complex post-translational pathways, including protein modifications, genetic codon preference, protein–protein interactions, chaperon-dependent conformation maintenance, degradation, etc. The effects of phosphorylation on the stability of circadian clock proteins are crucial for precisely determining protein function and turnover, and it has been proposed that the phosphorylation of core circadian clock proteins is tightly correlated with the circadian period. Nonetheless, recent studies have challenged this view. In this review, we summarize the research progress regarding the function, regulation, and mechanism of protein stability in the circadian clock systems of multiple model organisms, with an emphasis on *Neurospora crassa,* in which circadian mechanisms have been extensively investigated. Elucidation of the highly complex and dynamic regulation of protein stability in circadian clock networks would greatly benefit the integrated understanding of the function, regulation, and mechanism of protein stability in a wide spectrum of other biological processes.

## 1. Introduction

Living organisms have adapted to the daily rotation of the Earth on its axis. By means of cell-autonomous circadian clocks that can be synchronized to the daily and seasonal changes in external time cues, most notably light and temperature, organisms anticipate environmental transitions and perform activities during the day in sync with cycling environmental cues. Circadian rhythms are ubiquitous phenomena that recur daily in a self-sustaining, entrainable, and oscillatory manner with periods of approximately 24 h. Circadian clocks are critical timekeeping systems that control a wide spectrum of physiological and behavioral rhythms in most eukaryotic and prokaryotic organisms [1,2,3]. Circadian clocks regulate the expression of approximately 20–80% of protein-encoding genes in different species [4,5,6,7,8,9].

The rhythmic change in the abundance of core circadian clock proteins defines a critical nodal point for negative feedback and cause-and-effect oscillation within the circadian clock mechanism [1,10]; therefore, controlling the stability of circadian proteins is essential for maintaining robust and self-sustained oscillations.

## 2. The Molecular Network of the Circadian Clock

The Circadian clock controls the circadian rhythms of molecular, metabolic, physiological, and behavioral parameters within a period of approximately 24 h in most eukaryotes and certain prokaryotes [11]. “Circadian” means “about a day”, which derives from the Latin words “circa” and “diem”, respectively [12]. Circadian rhythms have several fundamental characteristic properties, which include the following: (1) persistability with a period of ~24 h under constant conditions without cycling external time cues, e.g., light or temperature; (2) resetability by external or internal cycling cues; and (3) temperature compensation which renders clock output largely insensitive to environmental temperature fluctuations [13].

At the molecular level, circadian clocks are controlled by a set of core circadian genes and many associated factors. The core circadian components constitute interlocked transcriptional and translational feedback loops, called transcriptional–translational negative feedback loops (TTFLs), which oscillate and drive the circadian expression of clock-controlled genes [10,14,15]. TTFL regulatory networks are present in all the identified kingdoms that possess circadian clocks, from prokaryotes to mammals (Figure 1).

Cyanobacteria emerged on Earth approximately 3–4 billion years ago, and some cyanobacterial species possess endogenous circadian clocks that control important processes, e.g., photosynthesis and nitrogen fixation [16]. The cyanobacterial circadian clock harbors both protein phosphorylation and transcription–translation cycles to produce circadian rhythms. In the unicellular cyanobacterium *Synechococcus elongatus* PCC 7942, KaiA, KaiB, and KaiC (KaiABC) are the core circadian components that constitute the pacemaker. KaiC has versatile activities, including autokinase, autophosphatase, and ATPase. KaiA binds to the A-loop of KaiC and promotes the phophosphorylation of S431 and T432 on KaiC, forming a post-translational oscillator (PTO) [17,18]. The circadian system in *S. elongatus* also contains TTFL, in which the transcripts from the *kaiBC* operon and the protein KaiC oscillate in a circadian manner in constant light. Overexpression of KaiC abolishes the expression of *kaiBC* [19]. The cyanobacterial PTO system can run independently of TTFL; however, the latter is necessary to stabilize and enhance the resetting of the Kai-based PTO in *Synechococcus* [20,21,22,23].

*N. crassa* is a filamentous fungus that has long served as an important circadian model organism in circadian research. The PAS domain-containing factors WHITE COLLAR 1 (WC-1) and WC-2 are circadian positive elements that form the heterodimer WHITE COLLAR complex (WCC). The WCC functions to activate the transcription of the *frequency* (*frq*) gene, which encodes the negative element of the circadian clock [24]. The FRQ-interacting RNA helicase (FRH) binds to FRQ and assists in the formation of the correct structure, proper phosphorylation, degradation, and association with other circadian and noncircadian factors [25,26,27]. The WCC progressively activates *frq* transcription before subjective dawn. The synthesis of FRQ is maximally repressed at subjective dusk, and the FRQ level decreases during the subjective night [28]. FRQ feeds back to repress the function of WCC as a transcription activator, thus ceasing its own transcription of *frq*. FRQ binds with casein kinase 1 (CSNK1, CK1) and other protein kinases, and hyperphosphorylated FRQ proteins undergo gradual degradation, after which WCC becomes free again and resumes its function in facilitating *frq* transcription. *Neurospora* TTFL directly controls the rhythmic expression of downstream clock-controlled genes (*ccgs*) by regulating the cis-elements located in the promoters of these genes, which are called Clock boxes (C-boxes) [10].

**Figure 1 ijms-25-02574-f001:**
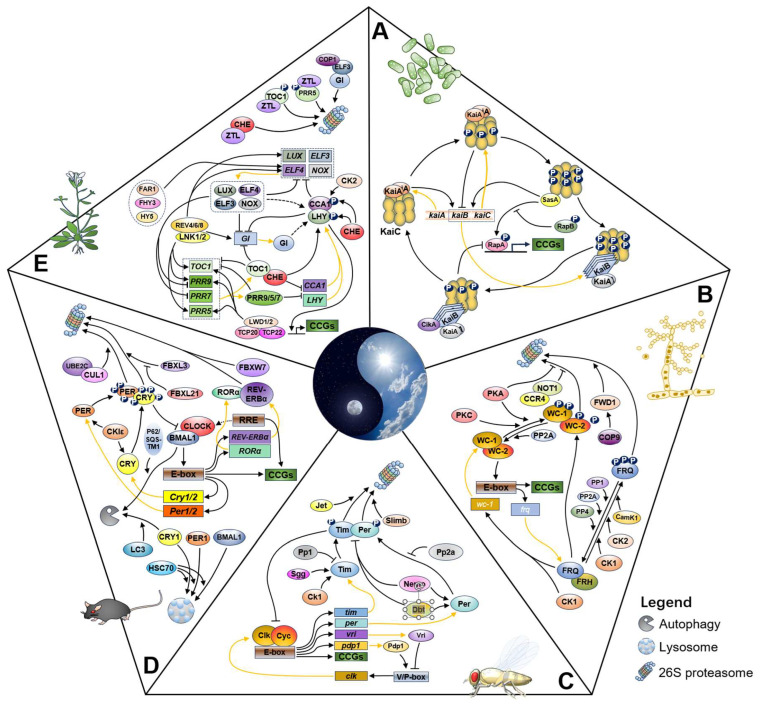
Circadian oscillators in different species. The circadian regulation in five important model organisms is shown, including cyanobacteria (*S. elongatus* PCC 7942) (**A**) [18,29], fungi (*N. crassa*) (**B**) [30], plants (*A. thaliana*) (**C**) [31,32,33], insects (*D. melanogaster*) (**D**) [11], and mammals (*M. musculus*) (**E**) [34]. Note that only the basic regulatory circuits are shown. Lines in orange highlight translation.

In *D. melanogaster*, the core oscillator proteins include the positive elements Clock (Clk) and Cycle (Cyc) and the negative elements Period (Per) and Timeless (Tim). Clk and Cyc function as transcriptional activators to control the rhythmic transcription of *per* and *tim*, which encode the negative circadian elements PER and Tim. From early to midnight, Per and Tim are predominantly localized in the cytoplasm and form heterodimers. When it is late at night, the Per/Tim complex enters the nucleus and represses the transcription activity of Clk/Cyc, which subsequently decreases the expression of *per* and *tim*. A series of post-translational modifications (PTMs) finely control the pace of accumulation and degradation of the Per and Tim proteins. The gradual degradation of TIM and then of PER removes the repression activity on Clk/Cyc and restarts the transcription of *per* and *tim*, and Clk and Cyc form heterodimers to activate the transcription of *per* and *tim* and other clock-controlled genes containing the *cis* element of the enhancer box (E-box) [35,36]. Furthermore, Vri and Pdp1 are leucine zipper motif-containing transcription factors, which regulate clock transcription through V/P-box; on the other hand, Clk and Cyc also control the expression of *vrille* (*vri*) and *par domain protein 1* (*pdp1*) through E-box elements in their promoters [37,38].

The mammalian circadian pacemaker is the suprachiasmatic nucleus (SCN) in the anterior of the hypothalamus [39]. The circadian rhythm is regulated by the positive transcription factors CLOCK and brain and muscle Arnt-like protein-1 (BMAL1) (also termed Arnt-like protein 1, ARNTL1), which induce the expression of the negative clock elements PER1/2 and CRY1/2, forming an important negative-feedback loop. PER2 is significantly more unstable than PER1, and the progressive phosphorylation of PER1 occurs significantly faster than that of PER2 [40]. The orphan nuclear receptors REV-ERBs (−α and −β) and retinoic acid receptor (RAR)-related orphan receptor (ROR) (−α, −β, and −γ) act in interconnected ancillary circuits that act as ancillary loops to regulate *Bmail1* expression. In this circuit, REV-ERB proteins inhibit, while ROR proteins activate, the transcription of *Bmal1* [41].

Plant circadian clocks endow plants with the advantageous ability to adapt to environmental alternations by controlling key essential processes, such as photosynthetic activity, hypocotyl elongation, and the floral transition [42,43]. In *A. thaliana*, the basic TTFL is composed of a pair of dawn-expressed MYB-related transcription factors, LATE ELONGATED HYPOCOTYL (LHY), and CIRCADIAN CLOCK ASSOCIATED 1 (CCA1), which are expressed around dawn. CCA1 and LHY homo- or heterodimerize and bind to the evening element (EE) motifs on the promoters of other clock components, including PSEUDO RESPONSE REGULATOR (PRR), TIMING OF CAB EXPRESSION 1 (TOC1/PRR1), GI, and LUX ARRHYTHMO (LUX), EARLY FLOWERING 3 (ELF3), and *ELF4,* which form the ELF3-ELF4-LUX evening complex (EC) and downstream genes, to repress the expression of those evening-phased genes.

In addition, LHY and CCA1 mutually repress the expression of these genes [32]. *PRR9*, *PRR7*, *PRR5*, *PRR3*, and *TOC1* are sequentially expressed and repress their own transcription, as well as that of *CCA1* and *LHY*, and the loss of these proteins alters the period. Moreover, the transcription of *PRR9* and *PRR7* is negatively regulated by the EC complex. The transcriptional coactivators LIGHT-REGULATED WD1 (LWD1) and LWD2 interact with the TEOSINTE BRANCHED 1-CYCLOIDEA-PCF20 (TCP20) and TCP22 to promote the expression of *CCA1*, *PRR9*, *PRR7*, and *TOC1* in the morning. In the afternoon, transcriptional activation is mediated by the NIGHT LIGHT-INDUCIBLE AND CLOCK-REGULATED proteins (LNK). RVE8 and RVE4 recruit transcriptional coactivators, and the RVE-LNK complex promotes the transcription of *PRR9*, *PRR5*, *TOC1*, *GI*, *LUX*, and *ELF4*. FHY3, FAR1, and HY5 are also implicated in the transcriptional activation of *ELF4*. In the evening, TOC1 represses all the daytime components, as well as *GI*, *LUX*, and *ELF4*. LUX and ELF4, together with ELF3, form the evening complex (EC), which is a transcriptional repressor of *GIGANTEA* (*GI*), *PRR9*, and *PRR7*. Moreover, GI and an EC variant containing BOA (NOX) function to activate the transcription of *CCA1* and *LHY* [31,32,33]. In addition to regulating CO transcription through the TTFL pathway, PRR proteins interact with and stabilize CO and facilitate its accumulation under long days, which increases the binding of CO to the promoter of FLOWERING LOCUS (FT) and consequently enhances FT transcription and early flowering [44].

Despite the evolutionary distance between clock genes in different species, the architectures of molecular clockworks are highly similar across kingdoms, and the regulation of the circadian clock is an example of convergent evolution [11,45]. In eukaryotes, all circadian systems involve progressive increases in the global phosphorylation of certain clock components until highly phosphorylated forms are targeted for rapid degradation [10,46]. Moreover, the regulators of post-translational modifications and degradation of circadian clock proteins are evolutionally conserved. For instance, CK1 in fungi and animals and CK2 in plants are crucial for the phosphorylation of circadian clock proteins. In eukaryotes, the ubiquitin–proteasome system and autophagy are involved in the modification and degradation of clock proteins [47,48].

## 3. The Regulation of Circadian Clock Protein Stability in the *Neurospora* Circadian Clock

*Neurospora* is an important model for circadian studies and has greatly propelled advances in the investigation of a variety of circadian mechanisms at the molecular level [30]. It has been proposed that fungal circadian clock systems are evolutionarily closely related to animal clocks [10].

### 3.1. Post-Translational Regulation of FRQ Protein Stability in Neurospora

FRQ is dynamically controlled by a large number of time-of-day-specific phosphorylation sites. The phosphorylation of FRQ determines its activities and binding partners and eventually leads to its inactivation (Figure 2) [49,50,51,52]. To measure the degradation rate of a circadian clock protein, cycloheximide (CHX) is usually added to inhibit translation, after which the cells/tissues are harvested at a series of time points and subsequently subjected to Western blot analysis and densimetric quantification. In *Neurospora*, the FRQ decay rate can also be observed in vivo by transferring *Neurospora* from LL to DD, which results in steady-state phosphorylation and degradation [50,52,53].

FRQ contains multiple degrons that are critical for PTM and FRQ turnover. Different blocks of the FRQ protein have different impacts on its own turnover [51,54]. Strains with mutations within the N’ 689 aa region of FRQ show decreased turnover rates of the FRQ protein and longer periods, but the strains with mutations within the region from aa 690–989 show increased FRQ turnover and shorter periods [50,51,55]. In some of the mutants harboring mutations at certain residues, e.g., FRQ1 (with the FRQ^G482S^ mutation), FRQ3 (with the FRQ^E364K^ mutation), FRQ7 (with the FRQ^G259D^ mutation), FRQ2 (with the FRQ^2895T^ mutation), and circadian periods are altered [56]. Among these mutants, FRQ7 is more stable than the wild-type control, while FRQ1 is less stable, which correlates with the circadian phenotypes [57].

FRQ has a large form of 989 aa (large FRQ, l-FRQ) and a small form of 890 aa (small FRQ, s-FRQ). l-FRQ has an extra 99 aa at the N’ terminus that is absent in s-FRQ. l-FRQ contains 132 serine and 56 threonine residues, 103 of which have been found to be phosphorylated in vivo or in vitro, and most of these sites undergo phosphorylation in a circadian fashion. Some of these sites are differentially phosphorylated between FRQ isoforms (Figure 2A) [50,51,52]. Baker et al. reported that an early-phase phosphorylated region can increase period length partially by stabilizing FRQ, while later phosphorylation of a physically separate region can decrease period length by decreasing FRQ stability [50]. In this scenario, the phosphorylation pattern of FRQ is similar to that of a drum in a music box, which has many convex dots, each presenting a note and touching the comb in a certain order to produce music (Figure 2B,C).

A strain exclusively expressing l-FRQ has a shorter period than a wild-type control; in contrast, a strain exclusively expressing s-FRQ has a longer circadian period [58]. The overall structure of l-FRQ is looser and more unstable than that of s-FRQ [52]. Similarly, loss of *Cry1* in mice causes a shorter circadian period, while deletion of *Cry2* causes a longer period [59].

CK1a is a major kinase involved in FRQ phosphorylation, but other kinases are also implicated in its phosphorylation. *Neurospora* FRQ harbors two CK1 binding regions, the FRQ-CK1 binding domain (FCD1, 319 aa–326 aa) and FCD2 (488 aa–495 aa), which may be rather close in the tertiary structure [55,60,61]. Deletion or mutation of the FCD1 and FCD2 regions abolishes the FRQ-CK1a interaction and CK1a-mediated FRQ phosphorylation [62]. Conidiation rhythmicity is abolished in a null strain of *pkac*-*1*, which encodes one of the catalytic subunits of cyclic AMP-dependent protein kinase A (PKA), and in a strain with a 19-bp deletion in the essential *pkar* gene, which encodes the regulatory subunit of PKA. PKA regulates the positive elements of the circadian clock by inhibiting the activities of WC proteins by serving as a priming kinase for casein kinases. In addition, PKA regulates FRQ phosphorylation and stabilizes FRQ [63]. These findings suggest that the crosstalk between different kinases plays a critical role in regulating FRQ phosphorylation and function.

**Figure 2 ijms-25-02574-f002:**
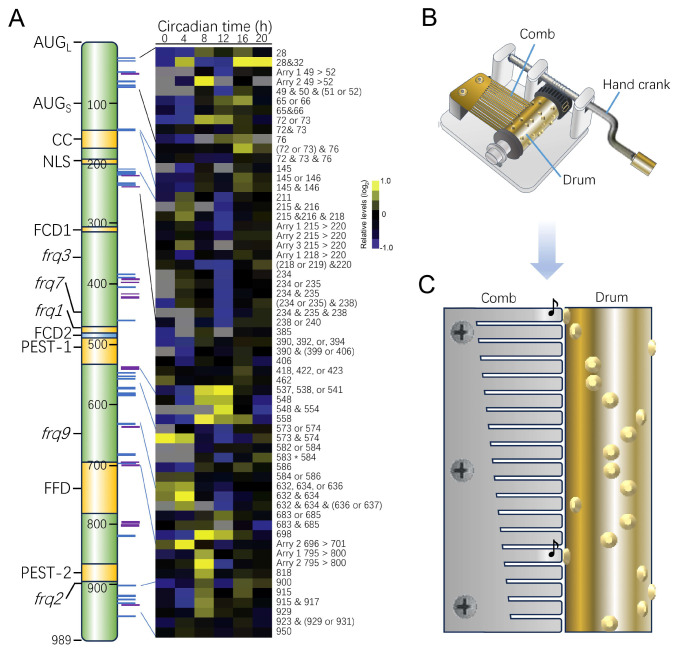
The regulation of FRQ phosphorylation. (**A**) FRQ contains more than 100 phosphorylation sites, a substantial set of which are phosphorylated in a circadian fashion [50,61]. (**B**,**C**) Internal structure of a music box (**B**) and the detailed structure of the comb and drum (**C**).

FRQ is an intrinsically disordered protein (IDP) that likely contains only a small amount of helical structure. FRH is a homolog of *Saccharomyces cerevisiae* Mtr4 [26,64]. FRH acts as a “nanny” molecule to stabilize the FRQ protein, although its RNA helicase activity is not necessary for circadian function. Disruption of the FRQ-FRH complex (FFC) interaction results in increased degradation of FRQ through the F-box and WD40 repeat-containing protein 1 (FWD-1)-dependent pathway and another unknown turnover pathway [25,27,65,66]. The highly unstructured N-terminal region (aa 1–150) of FRH is critical for FRQ-FRH association and FRQ stability [25,65].

### 3.2. Post-Translational Regulation of WC-1 and WC-2 Stability in Neurospora

The GATA-type zinc finger-containing WC-1 and WC-2 proteins are two PAS domain-containing transcription factors, of which WC-1 is a blue-light photoreceptor. WC-2 exhibits a constant expression pattern while WC-1 abundance oscillates with a low amplitude and a phase delayed by ∼6 h relative to that of FRQ [67,68]. WC-1 is less abundant than WC-2 and is the limiting factor of transcription factor capacity in the WCC [69]. WC-1 and WC-2 form heterodimeric complexes that activate *frq* transcription. The shuttling and activity of WCC are modulated by FRQ in a circadian fashion [28,70,71]. Multiple dispersed regions on FRQ, especially certain Gln/Glu residues in some of these regions, are necessary for its association with WCC. Deletion of these regions abolishes the interaction between FFC and WCC and decreases WCC levels. Mutations in some of these residues result in dramatically repressed FRQ levels owing to destabilization [49]. The WC-1/2 interaction is essential for maintaining the steady-state level of WC-1 and the function of WC proteins in the circadian clock and light responses [51].

The phosphorylation of WC proteins inhibits WCC as a transcription factor [55]. Despite their constant protein levels, the phosphorylation of WC-1/2 oscillates in a circadian manner [72,73]. WC-1 is phosphorylated through both FRQ-dependent and FRQ-independent pathways; the former is achieved by CK1 and CK2 recruited by FRQ, and the latter is achieved by additional kinases, e.g., PKA [74]. Phosphorylation of WC-1 at S885 and S887 by CK1 contributes to its accumulation, which is dependent on FRQ. FRQ promotes WC-2 expression at the transcriptional level [67,68,73]. Moreover, the abolition of the FRQ-FRH interaction leads to a decrease in WCC and increased WC-1 phosphorylation [27,75]. The abundance of nuclear FRQ is far less than that of nuclear WC-1; to close the negative feedback loop, the binding of FRQ to WC-1 in the nucleus leads to rapid phosphorylation and degradation of WC-1, which results in the inhibition of the transcriptional activity of WCC [76]. Negative feedback occurs in the nucleus, whereas positive support of WC-1 occurs in the cytosol, and positive and negative feedback loops are interlocked through FRQ-dependent phosphorylation of the WCC [28,73].

Mass spectrometry analysis revealed five in vivo phosphorylation sites in the WC-1 protein, including S988, S990, S992, S994, and S995. The degradation rate of the WC-1 protein did not significantly differ between a strain in which all five of these sites were mutated to alanine and the wild-type control, and mutations of S992 and S995 led to disruption of conidial circadian rhythms [77]. Sancar et al. identified S433 as the phosphorylation site on WC-2. In total, 80 and 15 phosphorylation sites were identified on WC1 and WC-2, respectively [78,79].

The mutual positive regulatory effects of FRQ, WC-1, and WC-2 are crucial for ensuring the robustness of circadian oscillation. Cytosolic FRQ proteins support WC-1 and WC-2 levels through different positive loops [28,70,71]. WC-1 is an inhibitor of *wc-2* transcription, and WC-1 is stable only in complex with WC-2; FRQ likely supports the accumulation of *wc-2* transcripts by repressing the activity of the WCC and may act indirectly as an activator of *wc-2* [28,67,68]. Likewise, such mechanisms function in the mammalian circuits of mutual regulation between CLOCK/BMAL1 and *Rev-erbα* and in the regulation of *vri* by Clk/Cyc in *Drosophila* [80].

FRH plays multiple roles in regulating the properties of FRQ [25,26,27,64,65,75,81]. The FRH^R806H^ mutation abolishes the FRQ-FRH interaction and circadian arrhythmicity; moreover, the protein levels of both WC-1 and WC-2 are repressed, and the degradation rate of WC-1 increases [75]. The loss of normal conformation and function due to the abolishment of the FRQ-FRH interaction may contribute to these alterations.

The CCR4–NOT complex is a conserved multisubunit functionally diverse machine that regulates gene expression [82]. NOT1 is a scaffold protein in this complex, and knockdown of NOT1 expression results in dramatically slowed growth and abolishment of circadian rhythms owing to decreased levels, hypophosphorylation, and increased degradation of WC proteins [74]. It is possible that NOT1 regulates WC degradation through association with other CCR4–NOT subunits, e.g., the E3 ubiquitin ligase CCR4 [82].

## 4. The Regulators of Clock Protein Stability in Different Organisms

PTMs are covalent, enzymatic, or nonenzymatic attachments of specific chemical groups to amino acid side chains that broadly control numerous important biological processes [83]. The fine control of protein stability is crucial for maintaining homeostasis and avoiding disease. The stability of core circadian clock components closely correlates with circadian homeostasis and determines circadian parameters. To date, a variety of different PTMs have been implicated in the control of the stability of circadian clock proteins [84].

### 4.1. Protein Degradation Pathways Implicated in the Circadian Clock

#### 4.1.1. Subsubsection

Regulated proteolysis is essential for normal physiology. In general, cellular proteins are degraded by two distinct but connected pathways: the ubiquitin–proteasome system (UPS), which eliminates short-lived proteins and soluble misfolded proteins, and the lysosome system, which eliminates long-lived proteins, insoluble protein aggregates, entire organelles, macromolecular compounds, and intracellular parasites via endocytosis, phagocytosis, or autophagy pathways [85].

Ubiquitin ligases, including Ub activating enzymes (E1), E2, and E3, attach 76-aa ubiquitin proteins to target proteins sequentially [85]. F-box proteins are key elements of Skp-cullin-F-box protein (SCF) complexes, a class of E3 ubiquitin ligase complexes that facilitate the ubiquitination of proteins for proteasomal degradation [86]. In *Drosophila*, Slimb, an F-box/WD40 repeat-containing protein, is a component of the SCF (Skp1/Cullin/F-box protein) ubiquitin ligase (E3), which mediates the ubiquitylation of the phosphorylated Per protein [87,88]. Per^S47^ phosphorylated by double-time (Dbt) can be recognized and bound to Slimb, which decreases Per stability and negatively correlates with circadian period length [89].

*Neurospora* FWD-1 is a homolog of Slimb that interacts with and ubiquitinates phosphorylated FRQ. FWD-1 inactivation leads to hyperphosphorylated, stabilized FRQ, and arrhythmic conidiation [66,77]. Nonetheless, the rapid decrease in FRQ due to the abolishment of FRQ-FRH interaction is independent of FWD-1, suggesting that the degradation of misfolded FRQ proteins may be mediated by a different pathway [27].

The COP9 signalosome (CSN) is a conserved protein complex composed of eight subunits (designated CSN1-8) that functions in the ubiquitin–proteasome pathway. COP9 mediates the deneddylation of the cullin subunit, which is required for recycling of the SCF^FWD−1^ complex [90]. Moreover, the conidiation of the *Neurospora* knockout strains *csn-1*, *csn-2*, *csn-4*, *csn-5*, *csn-6*, and *csn-7* is irregular. All these knockout strains showed dramatically lower expression of FWD-1 and several other E3 ubiquitin ligases [66,91]. In the *csn-5* knockout strain, FRQ protein is stabilized by repressing the FWD-1 level [92]. The stability of the SCF complex is regulated by the COP9 signalosome, which, therefore, has an indirect yet important function in the circadian system [77].

Plant cellular protein degradation is modulated by proteasome activity on ubiquitinated proteins in the cytosol, the autophagy of cytosolic complexes and organelles through vacuole-based protein degradation machinery, and a network of organelle-localized proteases [93]. TOC1 and PRR5 are potential proteolytic substrates of the E3 ubiquitin ligase SCF^ZTL^ (ZEITLUPE, ZTL) within the PRR family. The circadian photoreceptor and F-box protein ZTL associates with Skip and Cullin to form the Skip–Cullin–F-Box (SCF) E3 ligase complex, which ubiquitinates and targets TOC1, PRR5, and CCA1 HIKING EXPEDITION (CHE) for degradation by the 26S proteasome [94,95]. ZTL contains an N-terminal light–oxygen–voltage sensing domain (LOV) for sensing blue light, in addition to an F-box domain and a Kelch-repeat domain. The F-box domain endows ZTL with activity as an E3 ubiquitin ligase [96,97]. GI is a plant-specific nuclear protein with diverse functions in multiple physiological processes, including flowering time regulation, light signaling, hypocotyl elongation, and circadian rhythm control. A protein–protein interaction (PPI) between GI and ZTL stabilizes ZTL, and this interaction can be enhanced by light through the amino-terminal flavin-binding LOV domain of ZTL [98]. Two additional F-boxes containing ZTL family members, flavin binding, Kelch Repeat, F-Box1 (FKF1), and LOV Kelch Protein 2 (LKP2), are also involved in the regulation of the plant circadian clock [99]. Moreover, EARLY FLOWERING 3 (ELF3) interacts with MUT9-LIKE KINASE4 (MLK4), which is a granulin repeat family cysteine protease, and an F-box family protein (AT2G16365), which may be implicated in regulating ELF4 turnover [100].

CONSTITUTIVE PHOTOMORPHOGENIC 1 (COP1) is a RING-type E3 ubiquitin ligase that represses flowering by promoting CO degradation at night. ELF3 interacts with and recruits COP1 to increase the degradation of GI, which controls developmental responses, including the timing of flowering and seasonal changes in day length [101]. Moreover, the photoreceptor phytochrome (phy) A is also recognized and ubiquitinated by COP1 for degradation [102].

Ubiquitination and its counteracting process, deubiquitination, together contribute to the control of the circadian clock. Ubiquitin-specific peptidases (BIPs) (also known as ubiquitin-binding peptidases, UBPs) are the largest class of deubiquitylating enzymes DUBs in plants. UBP12 and UBP13, two members regulated by the circadian clock, are associated with ZTL, and reduced expression of UBP12 and UBP13 results in a longer period of expression [99]. Mutations in ubiquitin carboxyl Hydrolases 1, 2, and 3 (UCH 1/2/3), which are also members of DUBs, lead to prolonged periods of LHY and GI expression at high temperatures, probably by mediating their degradation [103].

In mammals, the F-Box protein family comprises approximately 40 members that direct ubiquitination and proteasome-mediated degradation of target proteins through the Spk/Cullin/F-Box protein (SCF) E3 ubiquitin ligase complex [104,105,106]. β-Transducin repeat-containing protein (βTrCP), also known as Fbxw1 or Fbxw11, is a homolog of Slimb and targets PER for degradation in a manner dependent on CK1-dependent PER phosphorylation. Phosphorylation of PER2^S478^ by CK1 recruits β-TrCP and facilitates PER2 degradation. Moreover, PER2^S478A^ knock-in mice show accumulated PER2 protein and longer locomotion periods [55,107,108]. In addition to SCFβ-TrCP, the ubiquitin ligase mouse double minute 2 homolog (MDM2) is involved in the control of PER2 degradation independent of PER2 phosphorylation [109].

Ubiquitin-binding enzyme 2C (UBE2C) facilitates PER1 degradation by promoting its ubiquitination via the E3 ligase S-phase kinase-associated protein 1 (SKP1)-cullin 1 (CUL1) [110]. The CRY1 and CRY2 proteins are ubiquitinated and degraded via the SCF^Fbxl3^ ubiquitin ligase complex, which is a prerequisite for the reactivation of CLOCK-BMAL1 and the consequent expression of *Per1* and *Per2* [105]. Mutations in FBXL3, e.g., C358S and I364T, stabilize CRY1 by blocking its proteasomal degradation and lengthen the circadian period [104,105,106].

Two competing E3 ubiquitin ligases are engaged in the control of the circadian period by degrading CRY. Loss of FBXL3 function leads to period lengthening, while mutation of F-box and leucine rich repeat protein 21 (Fbxl21) causes period shortening. FBXL21 plays dual roles: it protects CRY from FBXL3 degradation in the nucleus and promotes CRY degradation within the cytoplasm [111]. In the nucleus, FBXL21 counteracts the destabilizing effects of FBXL3 by ubiquitinating mouse CRY proteins and protecting them from FBXL3-dependent degradation [112].

REV-ERBα can be ubiquitinated and subsequently degraded by the F-box protein FBXW7; therefore, FBXW7 enhances clock gene transcription by repressing the inhibitory effects of REV-ERBα. Disruption of FBXW7 in the liver results in altered expression of core clock genes and perturbation of whole-body lipid and glucose metabolism. Phosphorylation of REV-ERBα^T275A^ by cyclin-dependent kinase 1 (CKD1) is necessary for its degradation by FBXW7. CDK1-mediated phosphorylation of REV-ERBα oscillates and contributes to rhythmic changes in REV-ERBα protein levels [113].

REGγ/PA28γ (encoded by the PSME3 gene) binds and activates the proteasome to promote ubiquitin- and ATP-independent cleavage of intact proteins and mediates protein degradation through a pathway different from the canonical ubiquitin–proteasome system [114]. REGγ KO mice exhibited shorter locomotion periods in constant darkness. Silencing or deficiency of REGγ caused elevated mRNA levels of *Per1*, *Per2*, *Cry1*, *Clock*, *Bmal1,* and *Rorα* and elevated protein levels of PER2 and BMAL1. REGγ directly promotes the proteasome-dependent degradation of BMAL1 protein, and GSK-3β can repress this process [115].

#### 4.1.2. Degradation of Circadian Clock Proteins Mediated by Chaperone-Mediated Autophagy

Cell homeostasis is dictated by the balance of the opposing biosynthesis and turnover. The lysosome is a cellular organelle for the degradation of target proteins with a wide array of resident acid hydrolases. In chaperone-mediated autophagy (CMA), cargo containing a peptide signal (KFERQ) binds with a cytosolic chaperone HSC70 and docks on the lysosomal surface receptor LAMP2A. HSCs facilitate the translocation of cargo into the lysosomal lumen [116].

CMA contributes to the rhythmic removal of clock machinery proteins, and disruption of lysosome-associated membrane protein type 2A (L2A) results in a phase shift and amplitude change in molecular rhythms. Conversely, circadian disruption abolishes the rhythms of CMA component expression and function. In addition to the protein turnover of clock components, the crosstalk between CMA and the circadian clock governs the oscillation of cellular proteome [117].

Macroautophagy is a highly conserved catabolic pathway responsible for quality control and degrades cytoplasmic contents in lysosomes. The mammalian circadian proteins BMAL1, CLOCK, REV-ERBα, and CRY1 are lysosomal targets, and macroautophagy affects the circadian clock by selectively degrading CRY1. CRY1 interacts with light chain 3 (LC3) through its LC3-interacting regions (LIRs) in autophagy. Two of the LIRs (^285^SLYGQL^290^ and ^492^SRYRGL^497^) are implicated in the regulation of CRY1 degradation and circadian control of hepatic gluconeogenesis [118]. BMAL1 inhibits ferroptosis through repressing *Egln2* transcription and consequently activating HIF1A [119]. CMA mediates the direct transport of substrate proteins containing the signaling peptide KFERQ to lysosomes for degradation [120]. A number of circadian proteins, e.g., BMAL1, CRY1, and PER1, contain the KFERQ signal and are degraded by CMA in lysosomes during the light or dark cycle, which is dependent on the canonical CMA pathway [117]. These findings demonstrate that several pathways are involved in the efficient removal of circadian proteins, which may function in a coordinated manner [116].

*Clock*^Δ19^ mice bearing a deletion of 51 amino acids in the CLOCK protein exhibit a very long periodicity of locomotion activity and a low amplitude under a constant environment [121]. The autophagy adaptor protein p62/SQSTM1 regulates CLOCK-induced BMAL1 degradation through autophagy, and CLOCK^Δ19^ mutation leads to stabilization of BMAL1 through attenuation of the proteasomal and autophagic degradation pathways [122].

In bacteria, the ubiquitin–proteasome system and lysosome are present and participate in the regulation of multiple aspects of infection and host defense in certain orders of bacteria [123,124,125]. However, the functions of these factors in circadian clocks have barely been investigated.

### 4.2. Dynamic Regulation of the Phosphorylation State of Circadian Clock Components

Dynamic phosphorylation of circadian clock proteins on a timescale is crucial for circadian timekeeping and regulates circadian rhythms through versatile roles ranging from regulating degradation to complex formation, subcellular localization, shuttling dynamics between the cytoplasm and nucleus, and activity [47,70,76,126,127,128].

All of the known circadian clock proteins are phosphoproteins [127]. Among the post-translational regulators, a set of protein kinases and phosphatases including CK1, CK2 and a calcium/calmodulin-dependent kinase (CAMK-1), PKA, checkpoint kinase 2 (Chk2), SHAGGY/GSK-3, cyclin-dependent kinase 1 (CDK1), and the phosphatases PP1, PP2, and PP4, are involved in the regulation of circadian clock [113,126,129,130,131,132,133,134,135].

CK1 and CK2 are also designated CKI and CKII, respectively, in some studies [28,60,136,137]. The CK1 family comprises a large number of conserved kinases in fungi, algae, and animals [138]. Mammals have six CK1 genes that encode CK1α, δ, ε, ɣ1, ɣ2, and ɣ3, along with several splice variants. CK1a forms homodimers, and its activity is regulated in multiple ways, including through inhibition of autophosphorylation and subcellular distribution [54,139,140].

The *Drosophila* core clock components, e.g., PER, are progressively phosphorylated and eventually degraded in a circadian manner to close the feedback loop [141,142]. PER has 25–30 phosphorylation sites, many of which are distributed in clusters [89,143]. The *Per^Long^* (*Per^L^*) mutant bears a V243D mutation with a period of 29 h; the *Per^Short^* (*per^S^*) fly mutant, which bears an S589N mutation, shows a ~19 h short period of locomotor rhythm. The *Per^0^* mutant bears a nonsense mutation at Q464, which is arrhythmic [144,145,146].

Similarly, a number of *Drosophila* kinases, including Dbt [141,147,148], Casein kinase 2 (Ck2) [149,150,151], Shaggy (Sgg/GSK3β) [152,153], and Nemo-like kinase (Nlk) [154,155], have been shown to modulate PER phosphorylation. *Drosophila* Dbt, an ortholog of human CK1ε, phosphorylates Per, which reduces its stability and the level of Per proteins. The *dbt^S^* mutants exhibited an 18.0 h short period, while the *dbt^L^* mutants exhibited a period of 26.8 h in constant darkness. The overt circadian rhythms are eliminated in *dbt^P^* mutants in which the expression of Dbt is abolished [141,147]. Dbt stably binds and phosphorylates PER and facilitates Per proteasome-dependent degradation in both the cytoplasm and nucleus, and phosphorylation by Dbt also controls the nuclear entry of Per [156,157,158]. The K38R mutation of Dbt still interacts with Per, but it lacks kinase activity, which causes an increased Per stability and a long circadian period [159].

In *Neurospora*, treatment with 6-dimethylaminopurine (6-DMAP), a general kinase inhibitor, blocks FRQ phosphorylation and stabilizes FRQ, suggesting the general effects of phosphorylation on the *Neurospora* circadian clock [160]. *Neurospora* Ck1a, a homolog of mammalian CK1ε/δ and *Drosophila* Dbt, is essential for viability and has a function in the circadian clock [10,108,161]. The FRQ-CK1a interaction is dynamic, and the affinity between these proteins decreases with increasing FRQ phosphorylation [55,81]. FRQ proteins are phosphorylated immediately upon synthesis [55]. The regulation of CK1 is also important for the control of the circadian clock. PERIOD 2 (PRD-2) participates in the regulation of the *Neurospora* circadian clock by stabilizing the *ck-1a* transcript, and the mutation of *prd-2* caused recessive inheritance of a long-period phenotype of ~26 h [162].

In *Arabidopsis*, the CASEIN KINASE 1 LIKE family (CKL) contains 13 CK1-like kinases. The functions of CKL kinases in regulating the circadian clock have not been fully elucidated compared to those of CK1 kinases in fungi and animals. Inhibition or knockdown of CKL lengthens the circadian period, likely through stabilizing PRR5 and TOC1 [138]. In addition, *Arabidopsis* has four MUT9-LIKE KINASEs (MLK, also known as PHOTOREGULATORY PROTEIN KINASE, PPK), which are close homologs of CKL members and interact with clock components [163].

CKIε is the mammalian homolog of *Drosophila* Dbt [164]. The semidominant autosomal *tau* mutation (CK1ε^R178C^) is a gain-of-function mutation in Syrian hamsters that causes significant shortening of activity cycles to 20 h in both SCN and peripheral tissues through the specific destabilization of nuclear and cytoplasmic PER proteins [46,165,166,167]. The CK1ε*^tau^* mutation promotes the degradation of both nuclear and cytoplasmic PERIOD but not of CRYPTOCHROME [168]. Nucleocytoplasmic partitioning of core clock components is essential for the proper orchestration and function of the circadian system [94]. In mice expressing mutated *Per2* that cannot interact with CK1, the phosphorylation of CLOCK is not induced. In these mice, PER proteins are stabilized and arrhythmically expressed, and the negative feedback loop is impaired [169].

The eukaryotic CK2 holoenzyme is an α2β2 heterotetramer [170], and *cka* is the only gene encoding the α subunit of CK2 in *Neurospora*. In the *Neurospora cka^RIP^* strain, in which CK2α is inactivated, FRQ is mostly hypophosphorylated, and its level is higher than that of the wild type. This strain shows dramatically slow growth and abolished rhythmicities in conidiation and *frq*/FRQ expression [137]. The phosphorylation is decreased, and FRQ degradation is decreased in this strain [171]. CK2 also contributes to the inactivation of the WCC and the regulation of temperature compensation [55,60,172,173].

Mammalian CK2 activity is involved in enhancing circadian amplitude and controlling the circadian period. CK2 binds with and phosphorylates PER2, and collaborates with CK1ε to promote PER2 degradation [174]. CK2α phosphorylates BMAL1 by maintaining the nuclear portion of BMAL1 [175]. CK2 subunits, including the regulatory subunits CKB3 and CKB4, play a critical role in the *Arabidopsis* circadian clock, and overexpression of these proteins leads to shorter circadian periods [176,177,178]. *Arabidopsis* CK2 interacts with LHY, CCA1, and phosphorylates CCA1 [179]. *Arabidopsis* CKB3 is associated with CCA1, and CKB3 overexpression leads to a shorter period [177].

In *Drosophila*, *shaggy* (*sgg*) encodes glycogen synthase kinase-3 (GSK-3), which promotes the nuclear transfer of PER/TIM complex through phosphorylating TIM. Elevation of *sgg* expression results in a shortened period length; in contrast, inhibition of GSK-3β leads to a shorter period in cultured mammalian cells [152,180].

Mammalian GSK-3β plays different roles in controlling the expression and function of clock proteins through the control of phosphorylation and degradation. GSK3βphosphorylates BMAL1 specifically on S17 and T21 and primes it for ubiquitylation. Inhibition of GSK3β activity enhances BMAL1 stability in the nucleus [181]. GSK3β phosphorylates BMAL1 specifically on S17 and T21, which primes it for ubiquitylation and proteasomal degradation, and inhibition of GSK3β leads to a shortened period length in mammalian cells [180,181]. Nucleocytoplasmic shuttling of BMAL1 plays an essential role in facilitating the accumulation of CLOCK in the nucleus and the degradation of the CLOCK/BMAL1 heterodimer. BMAL1 and CLOCK reciprocally induce each other’s turnover, while the binding of CRY stabilizes the CLOCK/BMAL1 complex. Transcriptional activation of CLOCK/BMAL1 is tightly associated with its turnover [182]. In addition, GSK3β promotes CRY2 degradation and the stabilization of Rev-erbα [183,184].

Spengler et al. identified a conserved phosphodegron containing a cluster of phosphorylation sites in CLOCK, ranging from aa 425 to 461. S431 in this region is a prerequisite phosphorylation site for the generation of BMAL-dependent phosphor-primed CLOCK and for potential GSK-3 phosphorylation at S427. Phosphorylation of these sites destabilized CLOCK proteins [185].

Piwi-like RNA-mediated gene silencing 1 (PIWIL1) and PIWIL2 participate in stem cell self-renewal, spermatogenesis, and RNA silencing. PIWIL1 and PIWIL2 play similar roles in activating the PI3K-AKT signaling pathway to phosphorylate and inactivate GSK3β and promote the stability of CLOCK and BMAL1 by repressing GSK3β-induced phosphorylation and ubiquitination of these two proteins in cancer cells. Mammalian GSK-3β is expressed in the suprachiasmatic nucleus and liver of mice and in NIH3T3 cells, and GSK-3β phosphorylation results in robust circadian oscillation [51,186].

Hyperphosphorylated PER has a more open structure and is more amenable to further global phosphorylation and degradation. In *Drosophila*, Dbt-dependent phosphorylation of PER at the N’ terminus promotes its binding to the F-box protein Slimb (β-TrCP), resulting in proteasome-dependent PER degradation [89].

In addition to its impact on the function of the circadian oscillator, the phosphorylation of clock proteins also modulates the expression of circadian output genes. The mTOR effector kinase, ribosomal S6 protein kinase 1 (S6K1), an important regulator of translation, rhythmically phosphorylates BMAL1. S6K1 phosphorylates BMAL1 at S42, and the phosphorylation at BMAL1^S42G^ stabilizes BMAL1 and increases its cytosolic distribution of BMAL1. S6K1 phosphorylation at BMAL1 contributes to the association with the translational machinery and stimulates protein translation of a set of genes [187].

The mammalian DNA-dependent protein kinase catalytic subunit (DNA-PKcs) interacts with CRY. Both repression of this kinase and phosphorylation at CRY1^S588^ by DNA-PKcs result in circadian rhythms with abnormally long periods, possibly through stabilization of the nuclear abundance and prevention of the FBXL3-mediated degradation of CRY1 proteins [188]. A set of phosphorylation sites in different circadian clock components and their associated circadian phenotypes in various organisms are listed in a review article [189].

In *Neurospora*, disruption of *ckb1* (encoding the catalytic subunit gene of CK2) results in hypophosphorylation and increased levels of the FRQ protein by decreasing its stability. The circadian rhythms of conidiation and the oscillation of FRQ occur over a long period with a decreased amplitude [137,171].

In addition to CK1, other protein kinases can be activated in response to external or internal cues and are implicated in the control of the circadian clock, although their effects on the circadian period are limited compared to CK1 [55]. In *Neurospora* and mammals, Chk2/PRD-4 interact and phosphorylate FRQ or PER1, respectively, in response to DNA damage and translational stress, triggering phosphorylation and destabilization [190,191]. Mammalian nutrient-responsive adenosine monophosphate-activated protein kinase (AMPK) phosphorylates and destabilizes CRY1. Treatment with the AMPK agonist aminoimidazole carboxamide ribonucleotide (AICAR) lengthened the circadian period and repressed the amplitude [192]. Other protein kinases are also involved in the regulation of circadian rhythms; however, their direct influences on the stability of circadian clock components remain unclear. In *Neurospora*, Ca/CaM-dependent kinase 1 (CAMK-1) and protein kinase A (PKA) may also regulate the circadian clock, although whether they phosphorylate FRQ in vivo remains undetermined [63,193]. It has been proposed that the phosphorylation of FRQ and WCC by kinases other than CK1 is not essential for the principal functions of the core oscillator [55]. In mice, cyclin-dependent kinase 5 (CDK5) phosphorylates CLOCK at residues T451 and T461, which accounts for the regulation of CLOCK stability and the cytoplasmic–nuclear distribution of CLOCK [194].

As a counteracting process, protein phosphatases also regulate the phosphorylation of circadian proteins. In *Neurospora*, the eukaryotic serine/threonine protein phosphatases PP1 (protein phosphatase 1), PP2A, and PP4 are involved in the circadian clock. PP1 and PP2A can dephosphorylate FRQ in vitro. The *Neurospora ppp-1* (catalytic subunit of PP1) mutant shows an advanced phase and a short period, and FRQ protein is unstable in this strain. In the *rgb-1* (a regulatory subunit of PP2A) mutant, FRQ stability is not affected, but the *frq* mRNA and FRQ protein levels are very low. This mutant has a low amplitude and a long period [195], suggesting that PP2A may regulate the circadian period through a pathway without direct impacts on FRQ stability.

Disruption of PP4 in *Neurospora* leads to a decreased stability of FRQ, a low amplitude, and a short period. Moreover, PP4 is required for the nuclear enrichment of WCC [47]. Consistently, in mice and flies, silencing of PP4R2, the regulatory subunit of PP4, shortens the circadian period, while overexpression of PP4R2 lengthens the circadian period [196]. *Drosophila* PER^S47^ is phosphorylated and destabilized by CKIδ and DBT in vitro, whereas PP2A (and/or PP1) stabilizes PER at least in part by decreasing the phospho-occupancy of PER^S47^ [89].

### 4.3. Preference for Genetic Codon Usage

Different organisms display different preferences for genetic codon usage; in one species, different genes exhibit genetic codon bias [197]. Intriguingly, the core circadian genes in the circadian systems exhibit overt codon bias in cyanobacteria, *Neurospora*, and *Drosophila*. The biased codon usage in circadian genes may be associated with specifically controlled translation rates [198,199,200,201,202].

*Neurospora frq* and *Drosophila Per* show dramatic preference for low-use codons, and their extensive optimization severely compromises or abolishes the circadian rhythms. Codon optimization causes less robust functions in promoting WC-1 and WC-2 levels in the circadian positive limb and changes in the stability and accessibility of the structure of the FRQ protein [199,202]. Similarly, the optimized codon usage of *Drosophila Per* leads to changes in the conformation, stability, and nuclear localization of Per in the pigment dispersing factor (PDF)-positive neurons in the brain [200]. Importantly, as many key circadian components across species are IDPs or contain large parts of IDP regions, regulation by phase separation may be an essential and conserved characteristic of circadian systems [25,203,204,205].

### 4.4. Additional Post-Translational Control Factors

Circadian clock components undergo complex PTMs; for instance, the expression and function of BMAL1 are mediated by a multitude of PTMs, including acetylation, phosphorylation, and protein instability by kinases, sumoylation, and ubiquitylation [115]. Many additional PTMs, e.g., acetylation and glycosylation, also regulate the stability of circadian clock proteins. The K560R, K576R, and K591R on CRY2 mutations abolish the acetylation of these three sites, and the mutated proteins are less stable. In contrast, in a mutant mimicking constant acetylation at these sites, the CRY protein showed an increased stability [206].

Covalent O-GlcNAcylation (O-GlcNAc) modifies the hydroxyl groups of Ser/Thr residues on proteins in addition to phosphorylation and exerts diverse functions, including protein–protein interactions, protein turnover, and subcellular localization [207]. *Drosophila* PER contains at least 6 O-GlcNAcylation sites, which undergo daily changes in O-GlcNAcylation. Furthermore, O-GlcNAcylation of PER leads to its stabilization [208,209]. O-GlcNAcylation at PER^S942^ attenuates the interaction between PER and CLK [209]. Inhibition of the O-GlcNAcylation pathway increases the period length in mammalian cells. The positive elements, BMAL1 and CLOCK, are rhythmically O-GlcNAcylated, which stabilizes BMAL1 and CLOCK by inhibiting their ubiquitination [210]. Phosphorylation and O-GlcNAcylation may compete on the same sites; for instance, the O-GlcNAc transferase and GSK3β have reciprocal regulations [211,212].

Overt circadian rhythms are absent in the plateau black-lipped pika. Epas1/HIF-2α is a transcription factor involved in the induction of genes regulated by oxygen. Epas-1 binds with BMAL1 at the E-box locus to initiate transcriptional activation. Pl-Pika *Epas1* has insertion mutations between exon 14 and exon 15, which disrupts an RNA splicing site. Under simulated hypoxic conditions, Pl-Pika-specific L-Epas1 proteins are significantly more stable than S-Epas1 present in other species, resulting in interference with circadian circuits and arrhythmic phenotype. L-Epas1-induced loss of circadian rhythms and enhancement against heart damage may be due to adaptations to the plateau environment [213].

Importantly, PPI between circadian components contributes to protein turnover. Mammalian PER3 interacts with PER1/2 and promotes nuclear translocation and stability [214,215]. *Drosophila* CLK acts to stabilize CYC, and loss of CLK results in low CYC levels [216]. In *Neurospora*, binding with WC-2 can stabilize WC-1 [69]. In *Neurospora*, the activated WCC as a transcription factor and LOV domain-containing protein VIVID (VVD) competes with the dimerization of the WCC, thereby reducing its activity and stabilizing WC proteins in photoadaptation [217]. The *Drosophila* scaffold protein, suppressor of Ras (SUR-8), interacts with PP1-87B to stabilize PER, and depletion of SUR-8 causes lengthening of the circadian period by approximately 2 h and delays TIM nuclear entry [218]. Moreover, the circadian clock regulates the stability of downstream proteins. For instance, the clock protein REV-ERBα directly interacts with O-GlcNAc transferase (OGT) and stabilizes OGT in different cellular compartments as the cellular localization of REV-ERBα oscillates [219].

Light exposure affects PTMs and the degradation of circadian proteins. In *Drosophila*, the newly translated PER proteins are required to maintain their stability and that of TIM proteins. Moreover, exposure to light increases TIM degradation [220,221,222,223]. Similarly, light exposure induces the accumulation of hyperphosphorylated WC-1 in an FRQ-independent manner in *Neurospora* [224]. In *Arabidopsis*, nuclear CO protein is degraded in the morning or in the dark, but light exposure stabilizes CO in the evening [225]. *Arabidopsis* CRYPTOCHROME 2 (CRY2) regulates light regulation of seedling development and photoperiodic flowering. CRY2 remains unphosphorylated, inactive, and stable in the absence of light; in contrast, blue light induces the phosphorylation of CRY2, photomorphogenic responses, and eventually degradation [226]. The *Arabidopsis* core clock protein GI associates with ZTL in a blue light-dependent manner, and the two proteins reciprocally stabilize each other. Their subsequent dissociation in darkness provides precise input to the photoperiod [98]. Ubiquitination also regulates light-induced responses through the photoreceptors CRYPTOCHROME 1 (CRY1) and CRY2. Upon light exposure, CRY1/2 proteins bind and activate CONSTITUTIVE PHOTOMORPHOGENESIS 1 (COP1), a RING domain E3 ligase. COP1, in conjunction with ELF3, destabilizes GI and tags it for degradation [227]. The day length also affects the stability of clock proteins (e.g., PRR5 degrades faster at LL19 and more slowly at LL7) [228].

Among the regulators of circadian clock protein stability, circadian regulation is involved in the expression, subcellular localization, or function of some of the regulators. For instance, mammalian CKIϵ and CKIδ and their interactions with circadian components show remarkable circadian rhythms in the nucleus despite the total protein levels being constant [148,229]. Consistently, the subcellular localization of *Drosophila* DBT in the nucleus but not the total DBT level oscillates in photoreceptor cells and lateral neurons [148]. The expression of plant CK2 is also under circadian control [178]. In the mouse liver, the activity and nuclear localization of AMPK oscillates, which is inversely correlated with CRY1 nuclear protein abundance [192]. The expression and phosphorylation of GSK-3β are rhythmically controlled robust circadian oscillations [129,230].

The expression of factors directly associated with protein decay, including some E3 ligases and the activity of the CMA-autophagy complex, displays overt circadian rhythms in a tissue-specific manner [117,160]. The expression of the *Arabidopsis* F-box protein ZTL is rhythmic, which controls the proteasome-dependent degradation of a central clock protein, TIMING OF CAB EXPRESSION 1 (TOC1), and is necessary for sustaining the circadian cycles [98,231,232]. The with-no-lysine (K) protein kinase (WNK) family members WNK1, WNK2, WNK4, and WNK6 are expressed in plants [233]. In the rat SCN, WNK3 binds with PER1 to regulate the circadian rhythms [234]. *Drosophila* TWINS (TWS), a PP2A regulatory subunit, is under circadian control [235].

Changes in clock protein stability and function have been studied only marginally in prokaryotes. In cyanobacteria, both the translation and degradation of KaiC are under circadian control. The synthesis rate of the KaiC protein reaches a peak around 12 h after exposure to LL (LL12), with a 6 h delay compared to the total KaiC level; the trough of degradation of KaiC occurs approximately 16 h after exposure to LL [236].

## 5. The Correlation of Clock Protein Decay with the Circadian Parameters

### 5.1. Fine Control of the Circadian Period

Accumulating evidence has demonstrated that PTMs, especially phosphorylation, play a very important role in controlling the circadian period. A set of protein kinases, including CK1, CK2, and protein kinase A, phosphorylate circadian components to regulate the stability of circadian elements and circadian period length [107,161,170,237]. It has long been proposed that the stability of FRQ protein is tightly correlated with the length of the circadian period [170,238]. Supportively, the mutant FRQ in the *frq^7^* mutant was degraded approximately twice as slowly as the corresponding wild-type control [237]. l-FRQ shows a faster degradation rate than s-FRQ, and consistently, the degradation period of the *l-frq* strain is shorter than that of the *s-frq* strain [52,58,239,240]. The *Neurospora ck-1a^L^* strain carrying a mutation equivalent to that of the *Drosophila dbt^L^* strain exhibited a ~32 h period due to increased FRQ stability. In this strain, the FRQ, WC-1, and WC-2 proteins undergo hypophosphorylation [60]. 6-DMAP treatment blocks FRQ phosphorylation, results in FRQ stabilization, and prolongs the period of the clock in a dose-dependent manner. The FRQ^S513^ mutation leads to a reduction in the FRQ degradation rate and a very long period (>30 h) [170].

Mutations in circadian negative components and CK1 in fungi and animals stabilizing these components lead to longer periods, and contrarily, those mutations destabilizing the circadian negative components lead to shorter periods [241]. The phosphotimer model has been proposed to explain the circadian clocks in multiple model organisms, in which a sequential series of phosphorylation events are regulated by multiple kinases at different sites over the course of a day. These phosphorylated sites interact and synergistically determine the stability of clock proteins and the length of the circadian period [107,143,242]. Phosphorylation regulates period length mainly through its effects on stability, together with other post-translational modifications. Protein degradation-associated factors are also determinants of circadian period length. For instance, ubiquitin ligase mouse double minute 2 (MDM2)-mediated PER2 turnover negatively correlates with the circadian period length in mammalian cells [109].

However, there are a variety of exceptions to the phosphorylation-stability-period model. Mutation of TIMELESS^R1081^ to a stop codon leads to stabilization of TIM, and mice bearing this mutant show a phase advance of sleep-wake behavior but no significant change in period length [243]. Different phosphorylation modifications multidirectionally contribute to the circadian period length, some of which accelerate the circadian pace, some deaccelerate, and some of which show no overt effects on the circadian pace. Phosphorylation on different regions on FRQ leads to opposing effects on circadian rhythms and FRQ stability, of which mutations of the phosphorylation sites at the N-terminal part result in lengthened periods through increasing FRQ stability although mutations of the C-terminal sites result in shorter periods due to decreased FRQ stability [50,51]. The FRQ stability does not always correlate with the circadian period length. The FRQ^T781A^ mutation confers a slower degradation of FRQ; however, there is no detectable change in the period length of circadian conidiation [52]. In addition, in contrast to most phosphorylation of FRQ by CK1 and CK2, FRQ phosphorylation by PKA increases FRQ stability [63].

In *Drosophila*, a number of hyperphosphorylation modifications are unrelated to PER stability, e.g., S48 modulates PER stability in a manner independent of phosphorylation at this site [89]. The PER^S942A^ mutant degraded at a similar rate as wild-type control in the presence of O-GlcNAc transferase, demonstrating that the S942A mutation has no significant effect on PER stability [209].

In circadian oscillators, phosphorylation has functions in addition to regulating the stability of circadian clock components, and these functions are closely interconnected [51]. For instance, the degradation of GSK-3β, a known promoter for BMAL1 degradation, exacerbates the effects of REGγ depletion in addition to its impacts on BMAL1 phosphorylation and control of its stability [115]. Dbt-dependent phosphorylation of *Drosophila* PER^S826,S828^, two sites located in a putative nuclear localization signal (NLS), can be rapidly induced by light and blocked by TIM. However, mutation of these two sites causes only mild changes in behavioral rhythms with a slightly longer period (by 1-h), suggesting that phosphorylation of these two sites may play an unknown role rather than controlling the circadian period [244]. Notably, some of the phosphorylation events show no impact on the circadian period, likely because some phosphodegrons only function as entire phosphorylated domains rather than as partially modified amino acids [53].

In *Neurospora fwd-1* and *csn-2* mutants, the circadian oscillator and output are decoupled, although the TTFL circuit is still functional, and the total clearance of the negative element is not an essential step. Despite the lack of rhythmicity in the bulk FRQ proteins and conidiation, robust rhythmicity of FRQ-LUC expression, which represents protein synthesis, persists in these strains. These findings suggest that FRQ stability is a correlative measurement but not a determinant of period length [53]. In some *Neurospora* strains, the FRQ-CK1 interaction, rather than FRQ stability, is correlated with the circadian period. A *Neurospora* strain with mutations in the Pro/Glu, (Asp)/Ser/Thr-rich sequence 1 (PEST-1) domain (M10) exhibits an extra long circadian period (32.1 h), although the turnover rate of FRQ is comparable to the wild-type control. Mutants with mutations in the FCD region, including FRQ^Q325N^, FRQ^Q494N^, FRQ^V320I^, and FRQ^L488V^, show overt associations between period length and the FRQ-CK1a interaction in these mutants [242]. These findings suggest that the FRQ-CK1 interaction plays more roles than just controlling the phosphorylation and stability of FRQ and that it modulates the circadian period through an undetermined mechanism rather than through a phosphorylation-stability-period circuit. Given that the FRQ-CK1 association is necessary for the hyperphosphorylation and inactivation of the WCC, it is likely that the altered WCC function may also account for the period length in these strains [60].

The stability of clock proteins closely correlates with the period length only within a certain range, and it may be decoupled. Based on the available knowledge, we infer that the stability of circadian proteins regulated by CK1 is the predominant determinant of circadian period length; however, the CK1-FRQ interaction also induces changes in the functions of other circadian clock components, which collectively participate in the determination of period length (Figure 3). For instance, it is possible that the PER2^S478A^ mutation may exert multiple functions more than the removal of phosphorylation at this site and influence the stability of PER2, e.g., affected subcellular distribution of PER1, CRY1, and CRY2 [107], which leads to superimposed effects on the circadian period.

**Figure 3 ijms-25-02574-f003:**
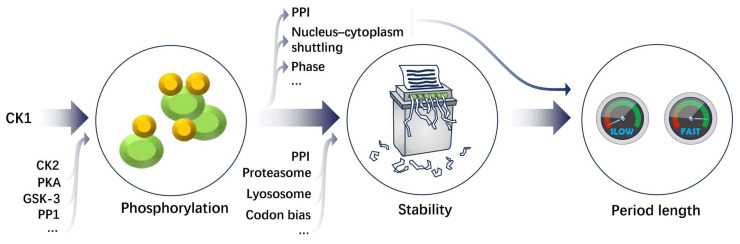
Numerous factors are involved in controlling the phosphorylation state, the stability of circadian components, and the length of circadian period at multiple layers. Protein homeostasis is one of the key regulatory mechanisms. There are many determinants of period length, among which CK1 is the master regulatory kinase of circadian clocks in different species [241]. The ellipses denote unlisted and unidentified factors or functions. PPI, nucleus–cytoplasm shuttling, and other regulatory mechanisms may also affect period length by modulating the PTMs and the turnover of clock proteins. The dashed line denotes the possible regulatory circuit to be confirmed.

Cyanobacterial core circadian components, KaiABC, dynamically interact with each other and can generate sustainable rhythms of KaiC phosphorylation in vitro [23]. Interestingly, the dynamics of the interaction of KaiA, KaiB, and KaiC determines the circadian period and amplitude of the in vitro oscillation [20], which demonstrates the importance of PPIs in determining the circadian period in prokaryotes.

### 5.2. Regulation of Circadian Temperature Compensation

Temperature compensation is a property of circadian rhythms that buffers the circadian period against the ambient temperature changes. An altered period length is usually associated with an abnormality in temperature compensation [238]. Hastings and Sweeney proposed that circadian temperature compensation may be a network mechanism arising from multiple counteracting reactions in response to temperature changes that cancel out their opposing effects on the period length [155,245].

The *Neurospora* mutants of *ck2β1* and *ck2α* exhibit abnormal temperature compensation features compared to those of the wild-type strain. Mutations in the CK2 targeting sites in FRQ phenocopied the deficient temperature compensation in the *ck2β1* and *ck2α* mutants. These data demonstrate that CK2 is involved in temperature compensation [172]. The *Neurospora* strain with a mutation in *prd-4*/*chk2* has a short period and defective temperature compensation, but the turnover rate of FRQ has not been determined [246]. In *Neurospora*, prediction with the Goodwin model suggested that the stability of FRQ should determine the length of the circadian period as well as temperature compensation [57]. Moreover, the CK1-FRQ interaction not only determines the circadian period but also contributes to temperature compensation [62].

The degradation rate of PER2 regulated by CK1ε/δ-dependent phosphorylation is temperature-insensitive in living clock cells but sensitive to chemical perturbations [247]. Zhou et al. proposed that some phosphorylation sites in PER2 may compete with each other to promote or repress PER2 stability. This regulation is temperature sensitive, i.e., PER2 degradation decreases at higher temperatures and increases at lower temperatures [248].

The circadian period length closely correlates with its phase [249], and some examples are presented here. Inhibition of CK1ε by PF-670462 treatment in mammalian cells results in a lengthened period and a delay in the circadian phase [250]. CK1 phosphorylation of PER2^S478^ recruits the ubiquitin E3 ligase β-TrCP, which leads to accelerated PER2 degradation. Moreover, PER2^S478A^ mutant mice exhibited a longer period and a delayed phase in the liver [107]. CAMK-1 also affects the circadian phase in *Neurospora*, leading to a delayed phase but with no significant effects on the circadian period [193]. The *Neurospora ck-1a^L^* strain also exhibited a significant phase delay in conidial rhythms and the FRQ progressive phosphorylation as well [60]. The mammalian protein kinase C (PKC) family is involved in regulating circadian phases. The neuron-specific protein PKCγ participates in food entrainment of circadian rhythms by partially stabilizing and reducing the ubiquitylation of BMAL1 [251]. GSK-3β overexpression advances the phase of clock gene expression; in contrast, inhibition of GSK-3β causes a delay in the circadian phase [129]. Among the properties of circadian rhythms, we emphasize the circadian period, while the circadian phase and temperature compensation are only briefly mentioned due to space limitations.

## 6. Abnormal Turnover of Circadian Clock Proteins and Disorders

Desynchronization of the circadian clock compromises fitness to environmental cues [2,252], and in humans, circadian misalignment has been associated with numerous disorders, e.g., sleep and mood disorders [253]. Misalignment in the circadian clock is usually correlated with sleep disturbance, and it has been demonstrated that some of the core clock genes (*PER2*, *PER3*, *Casein kinase* Iδ, *CRY2*, *DEC2*) affect the timing of sleep-wake behavior or the sleep duration in humans. Patients with familial advanced sleep phase syndrome (FASPS) exhibit a dramatic early sleep phase that occurs 4–5 h earlier than in unaffected individuals. The PER2^S659G^ and PER2^S662G^ mutations are associated with FASPS, which promotes the degradation of PER2. The accelerated degradation of PER2 caused by PER2^S569G^ and PER2^S662A^ mutations resulted in shorter periods [254,255,256,257], which is consistent with the positive relationship between circadian period length and phase [258]. The PER3^P415A/H417R^ variant, which is associated with FASPS and seasonal mood traits, destabilizes PER3 and PER1/2 [215]. Flavin adenine dinucleotide (FAD) is a mediator of the circadian clock through stabilizing CRY proteins, and CRY2^A260T^ mutation, a missense mutation localized in the (FAD) binding pocket, causes an advanced sleep phase in humans. The turnover rate of CRY2^A260T^ is significantly faster in both the nucleus and the cytosol [259,260].

The expression of a substantial set of genes associated with disease or drug development is under circadian control [8]. Given the important roles of protein kinases in the circadian clock, it is not surprising that pharmacological attempts to manipulate circadian rhythms have identified many molecules targeting these kinases. Hirota et al. identified a small molecule named longdaysin, which is effective at lengthening the circadian period in a dose-dependent manner. Longdaysin targets several protein kinases that are crucial for the circadian clock, including CKIδ, CKIα, and ERK2. Longdaysin treatment inhibits PER1 degradation, which causes a longer period through the combined effects of these kinases [261]. N-(3-(9H-carbazol-9-yl)-2-hydroxypropyl)-N-(furan-2-ylmethyl)methanesulfonamide (KL001), a small CRY-interacting molecule, can prevent ubiquitin-dependent degradation of CRY1 and CRY2 and thus elicits a lengthened circadian period [262]. In a large-scale screening of small compounds, three compounds that inhibit CK1ε activity and lengthen the circadian period were identified [263]. Isojima et al. identified pharmacologically active compounds that can inhibit the phosphorylation of PER2 by CKIε or CKIδ, stabilize PER2, and lengthen the circadian period as a consequence [247].

Abnormalities in circadian rhythms correlate with bipolar disorder (BD), seasonal affective disorder (SAD), and major depressive disorder (MDD) [264,265,266,267]. As a mood stabilizer, lithium is an effective treatment for BD and major depressive disorder [267,268]. Lithium treatment improves sleep and circadian rhythms. Lithium is a GSK-3 inhibitor that can alter the stability and/or nuclear translocation of clock proteins, including CLOCK, CRY2, PER2, and REV-ERBα, and treatment with lithium causes phase delay and/or period lengthening of biochemical and physiological variables [268,269,270]. The effects of lithium on circadian rhythms are well conserved in a variety of species; for instance, in *Neurospora*, treatment with lithium led to increased FRQ protein stability, prolonged period lengths in a dose-dependent manner, and partial loss of temperature compensation [271].

## 7. Prospects

The circadian period and temperature compensation are controlled at multiple post-translational layers, similar to a nesting doll. It is necessary to carry out systemic profiling experiments to identify additional PTMs controlling the stability of circadian clock components, which would help elucidate their functions and mechanisms. The correlation between CK1 and the circadian negative component and period length will be important for validating the findings in *Neurospora* in other species.

To date, many kinases and kinase regulators have been revealed to be potential targets for adjusting circadian rhythms, and an increasing understanding of the detailed regulatory mechanisms of clock protein stability will highlight the implications for the development of new therapeutics. As many of the regulators of protein stability have pleiotropic effects in addition to affecting the circadian clock, in the future, to be effective and practicable, identifying and developing small molecules or drugs to specifically mitigate circadian disorders without affecting other important biological processes is crucial.

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
