# Peer review of "The Function, Regulation, and Mechanism of Protein Turnover in Circadian Systems in Neurospora and Other Species"

_ijms, 2024, doi:10.3390/ijms25052574_

Round 1
Reviewer 1 Report
Comments and Suggestions for Authors
This is a well-composed review of comprehensive literature on post-translational modification and its implications for the circadian period, with casein kinase highlighted as a central mediator between the two, and as an evolutionarily preserved component across species. Their analogy of the drum in the music box (ll. 192-195 & Fig. 2) is especially inspiring and can be further expanded to explain inconsistencies between phosphorylation and period length. The criticism that FRQ stability does not always correlate with period length (ll. 729-730) can be extended to many more examples in the mammalian system in this context. Overall, this review is beneficial to the field. I only have minor comments for the improvement of the text.
Minor comments:
l. 70: independentlu -> independently (typo)
l. 465: homologue -> homolog, or consider ortholog. which is technically more proper.
ll. 512-512 and ll. 512-513 are repeated sentences.
l. 545: DNA: remove boldface.
l. 547: by DNA: remove boldface.
l. 577: phosphorylate -> dephosphorylate
l. 645: OGT -> O-GlcNAc transferase (OGT)
ll. 654-655 & ll. 655-656: sentence repeated (remove one).
l. 686: marginallyin -> marginally in
l. 688: LL12 -> 12 h after exposure to LL (LL12)
l. 690: 16 h after exposure to LL -> LL16
l. 632: PER2, PER3, ... -> Per2, Per3, (lowercase Italic for transcripts); and remove website links embedded in text.
l. 633: sleep -> remove website link.
Author Response
Point-to-point replies to reviewers’ comments
Reviewer #1:
Re:You are correct. We have revised independently in line 70. And we have rewrited homologue in line and delete the repeated sentences in lines 512-513 and 654-656. The surface of 'DNA' in line 545, 547 has been changed as required. We have revised phosphorylate to dephosphorylate. And we have added the full name of OGT in line 645. Blank space between marginally and in has been added in line 686. Two sentences in line 688 and 690 has been revised. We have removed website link and changed the PER2 and PER3 surface in line 832-833.
- 70: independentlu -> independently (typo)
Re: We have corrected this typo (pp2, line: 70).
- 465: homologue -> homolog, or consider ortholog. which is technically more proper.
Re: We changed to use homolog, as it may have several counterparts, which may contain both orthologs and paralogs (pp11, line 465).
- 512-512 and ll. 512-513 are repeated sentences.
Re: We have removed one of the repeated sentences (pp11, lines 509-510). And we adjusted the order of cited references 187-189 accordingly in this paragraph (pp12, lines 518-519; pp26, lines 1323-1328).
- 545: DNA: remove boldface.
Re: We have corrected its format (pp12, line 544).
- 547: by DNA: remove boldface.
Re: We have corrected its format (pp12, line 545).
- 577: phosphorylate -> dephosphorylate
Re: We have corrected this error (pp13, line 576).
- 645: OGT -> O-GlcNAc transferase (OGT)
Re: We have revised it accordingly (pp14, lines 644-645).
- 654-655 & ll. 655-656: sentence repeated (remove one).
Re: We have removed one of the repeated sentences (pp14, lines 653-654).
- 686: marginallyin -> marginally in
Re: We have added space between these two words (pp15, line 684).
- 688: LL12 -> 12 h after exposure to LL (LL12)
Re: We have revised it by following the comments (pp15, lines 686-687).
- 690: 16 h after exposure to LL -> LL16
Re: We have revised it by following the comments (pp15, line 688).
- 632: PER2, PER3, ... -> Per2, Per3, (lowercase Italic for transcripts); and remove website links embedded in text.
Re: Since here these clock genes denote those transcripts in human, so all the letters are in italic and capital. The website link has been removed (pp18, line 832).
- 633: sleep -> remove website link.
Re: We have conducted the corrections by following the comments (pp18, line 833).

Reviewer 2 Report
Comments and Suggestions for Authors
Circadian clocks take important role in the physiological and behavioral activities of almost all organisms. Its mysterious mechanism has attracted many researchers. The 2017 Nobel Prize in Physiology or Medicine was awarded to Jeffrey C. Hall, Michael Rosbash and Michael W. Young for their their discoveries of molecular mechanisms controlling the circadian rhythm. However, the mechanism is still uncovered. This review article summarizes the research progress on the function, regulation and mechanism of protein stability in the circadian clock systems, especially focusing on that of Neurospora. Since this is a very comprehensive review covering almost all recent studies, it will be informative to the researchers. However, it is difficult to understand the mechanism how protein turnover regulate the circadian clock. Figure 3 is too abstract. The authors should give an illustration to show their idea on how protein turnover regulates the circadian in Neurospora.
Comments on the Quality of English LanguageNone
Author Response
Point-to-point replies to reviewers’ comments
Reviewer #2:
Circadian clocks take important role in the physiological and behavioral activities of almost all organisms. Its mysterious mechanism has attracted many researchers. The 2017 Nobel Prize in Physiology or Medicine was awarded to Jeffrey C. Hall, Michael Rosbash and Michael W. Young for their discoveries of molecular mechanisms controlling the circadian rhythm. However, the mechanism is still uncovered. This review article summarizes the research progress on the function, regulation and mechanism of protein stability in the circadian clock systems, especially focusing on that of Neurospora. Since this is a very comprehensive review covering almost all recent studies, it will be informative to the researchers. However, it is difficult to understand the mechanism how protein turnover regulates the circadian clock. Figure 3 is too abstract. The authors should give an illustration to show their idea on how protein turnover regulates the circadian in Neurospora.
Re: We have rephrased the second paragraph in pp17, and provided more details in describing the findings in reference 110, which may help in understand Fig. 3. Overall, although the CK1-FRQ interaction is critical for determining the circadian period in some Neurospora strain, we think that the finely controlled phosphorylation and degradation still play a stem role in regulating the circadian period. Meanwhile, other factors are also involved. Therefore, The CK1-FRQ interaction and phosphorylation-controlled degradation are coupled in most cases, but in some cases, they may be decoupled.
The revised paragraph is “The stability of clock proteins closely correlates with the period length only within a certain range, and it may be decoupled. Based on the available knowledge, we infer that the stability of circadian proteins regulated by CK1 is the predominant determinant of circadian period length; however, the CK1-FRQ interaction also induces changes in the functions of other circadian clock components, which collectively participate in the de-termination of period length (Fig. 3). For instance, it is possible that the PER2S478A mutation may exert multiple functions more than the removal of phosphorylation at this site and influence of the stability of PER2, e.g., affected subcellular distribution of PER1, CRY1, and CRY2 [110], which leads to superimposed effects on the circadian period”.

Reviewer 3 Report
Comments and Suggestions for Authors
This is an excelent review embracing different experimental systems. I wuld expect many circadian biologists would enjoy reading this review on complex systems from generalist's point of view. Congratlations!

Author Response
Re: Thanks for your comments. We have also made revisions according to other reviewers' comments.